# Lunarminer Framework for Nature-Inspired Swarm Robotics in Lunar Water Ice Extraction

**DOI:** 10.3390/biomimetics9110680

**Published:** 2024-11-07

**Authors:** Joven Tan, Noune Melkoumian, David Harvey, Rini Akmeliawati

**Affiliations:** 1Discipline of Mining and Petroleum Engineering, School of Chemical Engineering, The University of Adelaide, Adelaide 5005, Australia; 2School of Electrical and Mechanical Engineering, The University of Adelaide, Adelaide 5005, Australia

**Keywords:** off-earth mining, nature-inspired behavior, water ice extraction, Lunarminer, biomimetics, bio-inspired robot, robot mechanism, swarm robotics, robot control

## Abstract

The Lunarminer framework explores the use of biomimetic swarm robotics, inspired by the division of labor in leafcutter ants and the synchronized flashing of fireflies, to enhance lunar water ice extraction. Simulations of water ice extraction within Shackleton Crater showed that the framework may improve task allocation, by reducing the extraction time by up to 40% and energy consumption by 31% in scenarios with high ore block quantities. This system, capable of producing up to 181 L of water per day from excavated regolith with a conversion efficiency of 0.8, may allow for supporting up to eighteen crew members. It has demonstrated robust fault tolerance and sustained operational efficiency, even for a 20% robot failure rate. The framework may help to address key challenges in lunar resource extraction, particularly in the permanently shadowed regions. To refine the proposed strategies, it is recommended that further studies be conducted on their large-scale applications in space mining operations at the Extraterrestrial Environmental Simulation (EXTERRES) laboratory at the University of Adelaide.

## 1. Introduction

The Moon has become the focus of extraterrestrial exploration due to its rich mineral resources (iron, titanium, uranium, rare earth elements, and water ice in permanently shadowed craters) [1]. These resources are essential for long-term crewed missions and for producing propellants through in situ resource utilization (ISRU) for deep space missions [2]. Lunar exploration began with the Luna program conducted by the Soviet Union, and NASA’s Apollo 11 mission to the Moon in 1969 was a landmark step in human space exploration [3,4]. The 21st century missions, such as NASA’s Lunar Reconnaissance Orbiter (LRO), have confirmed the presence of water ice at lunar poles, highlighting the Moon’s potential as a launch pad for deep space missions [5,6]. In addition, the lunar regolith contains helium-3 and rare earth elements, which are essential for clean energy and cutting-edge technologies [7].

The Moon holds great potential, but also presents challenges due to its harsh environment. Extreme temperatures, vacuum, and abrasive regolith are factors that have caused wear, tear, and technical difficulties for Apollo missions and China’s Yutu lunar rover [8,9]. The high cost of transporting water to the lunar surface, estimated at USD 35k to USD 70k per kilogram [10], further complicates lunar exploration. The rover technology available today is not suitable for extensive space mining [8]. The ISRU research has made progress in areas such as oxygen production, water ice extraction, and habitat construction; however, deep excavation and large-scale space mining remain underexplored [6]. The difficult terrain, communication issues, extreme temperature fluctuations, abrasive regolith, and low gravity on the Moon make terrestrial mining technologies unsuitable for applications on it [11,12].

Numerous technologies have been developed for the extraction of lunar water ice including NASA’s mobile in situ water extractor (MISWE) and the regolith advanced surface systems operation robot (RASSOR). However, these technologies are susceptible to failure, due to their high energy consumption and reliance on single large units. For example, MISWE’s extraction rate of 4.8 L per day with 12 units is much less than the proposed Lunarminer framework’s potential of 181 L per day, which is enhanced by a distribution of smaller robots that are optimized for fault tolerance and energy efficiency [13]. A similar drum-based excavation system is used by NASA’s RASSOR, which depends on constant operation and is energy-intensive and prone to mechanical failures [14]. In contrast, the Lunarminer framework uses biomimetic swarm robotics which are motivated by the synchronized flashing behavior of fireflies and the division of labor seen in leafcutter ants. This allows for improved fault tolerance, energy efficiency, and adaptability in the demanding lunar environment. Opportunity, Sojourner, and Curiosity are examples of Mars’s exploration rovers that have shown limitations in harsh environments, due to mechanical failures and mobility problems over time [15,16,17]. On the other hand, the Lunarminer’s distributed swarm methodology provides improved adaptability and resilience for mining activities on challenging lunar terrain. Therefore, the Lunarminer framework may be a good fit for future ISRU missions, since it provides a more effective, scalable, and fault-tolerant method of extracting lunar water ice.

The proposed Lunarminer framework investigates the application of swarm robots in lunar mining. The framework uses biomimicry, drawing inspiration from the collective behavior of insects and animals, to develop a coordination method for collaboration, decision-making, and task execution among swarm robots. This framework developed for lunar mining may have the potential to transform the terrestrial mining industry by improving its efficiency, sustainability, automation, and safety.

## 2. Related Work

The Lunarminer framework aims to contribute to enhancing ISRU missions, particularly in water ice extraction, by employing a bio-inspired approach to autonomous resource extraction. This framework draws inspiration from the division of labor observed in leafcutter ants for efficient task allocation and utilizes synchronized flashing behavior from fireflies for fault tolerance and recruitment.

### 2.1. State-of-the-Art in ISRU Technology

As space exploration progresses, ISRU becomes more essential for reducing the reliance on Earth’s resources. ISRU seeks to use resources from Mars and the Moon to produce fuel, oxygen, and water [18]. Significant progress has been made in extracting water and oxygen from the regolith and lunar ice, with the development of technologies such as carbothermal reduction and molten salt electrolysis [19]. The Moon is an ideal place for a space refueling station, because its regolith contains 40% oxygen [20]. Currently, water ice has been discovered in the permanently shadowed regions (PSRs) of the Moon, creating more opportunities for ISRU. NASA uses laser technology to map ice deposits, such as Lunar Flashlight and LunaH-MAP [21]. Future missions, such as NASA’s Lunar Crater Observing and Sensing Satellite (LCROSS) and the Volatiles Investigating Polar Exploration Rover (VIPER), are expected to provide important information about water and volatiles present in the lunar regolith [22]. Extraction methods such as sublimation and microwave heating are currently being developed [23]. If water can be efficiently extracted from the lunar ice, thus reducing the need for Earth-based supplies, more sustainable long-term crewed missions could become possible [24]. ISRU is also investigating regolith-based in-space manufacturing, particularly through 3D printing, to produce building materials and spacecraft components from lunar or Martian regolith [23,25]. For example, mixing magnesium oxide with lunar regolith and solidifying it with binding salts could produce printable materials for lunar habitats [26,27]. There are still risks in designing reliable systems for the harsh lunar environment, such as extreme temperature fluctuations, abrasive regolith, and high energy demands in the PSR [19]. To address these issues, NASA’s Lunar Surface Innovation Initiative is focusing on long-term resource extraction and construction technologies [28].

### 2.2. Comparative Analysis of ISRU Technologies and the Lunarminer Framework

With its decentralized swarm-based methodology, the Lunarminer framework enhances scalability, fault tolerance, and energy efficiency over other current ISRU technologies. NASA’s RASSOR system [14], for instance, is vulnerable to mechanical failure, because it depends on continuous operation and employs a drum-based excavation technique. Lunarminer, in contrast, employs several small robots, each of which performs a specific task that is modeled after the leafcutter ant’s division of labor. Because the system is not compromised by the failure of a single robot, this allows for more resilient operations and dynamic resource allocation. Comparably, the MISWE system [13] from NASA also has issues with relatively low extraction rates and high energy consumption. The MISWE is intended to extract water from lunar regolith with a single unit; however, with 12 units, it only produces 4.8 L per day. By providing a more scalable solution, Lunarminer’s bio-inspired swarm of robots surpasses the MISWE, in terms of daily water production reaching up to 181 L. Reduced downtime and greater adaptability to the lunar environment result from the decentralized approach and increased flexibility in resource allocation. Current rover-based technologies like Opportunity [15], Sojourner [16], and Curiosity [17] have advanced our understanding of planetary exploration, but they are not designed to extract large amounts of resources. Over time, these rovers have experienced mechanical wear and restricted mobility, especially in harsh environments like on the Moon. Furthermore, because of their larger mechanical components, rovers require more energy compared to Lunarminer, which distributes tasks among smaller and more efficient robots to minimize energy consumption. Also, Lunarminer’s dispersed swarms of smaller robots are designed for lunar mining tasks, can adapt to harsh environments, and can lessen the effects of individual robot failures. By addressing the shortcomings of the conventional systems, the Lunarminer framework may advance the ISRU’s objectives for resource extraction, making it more independent, effective, and scalable.

## 3. Theoretical Framework

This study aims to develop a swarm robotic system for lunar water ice extraction. Key challenges included creating a simulated lunar environment and designing robotic swarms. The Robot Operating System (ROS) was chosen for its scalability and adaptability, enabling the development of a complex system where robots collaborate, communicate, navigate, and make decisions to extract water ice under lunar conditions.

### 3.1. Mining Site Selection: Case Study of Shackleton Crater

Lunar craters, e.g., Shackleton Crater, located at the lunar poles, are considered to have a high probability of forming permanent ice at the crater bottom, with permanently shadowed areas acting as “cold traps” capturing volatiles such as hydrogen in the form of water ice [29]. The age of Shackleton Crater ranges from 1.1 billion to 3.3 billion years, belonging to the Eratosthenes period [30], and the crater has a diameter of 21 km and a depth of 4.2 km [31]. The crater walls have an average slope of 31°, and the floor diameter is 6.8 km [32]. The crater floor is flat [33], with minimal thermal fluctuations [34]. The frozen lunar regolith in Shackleton Crater, situated at the lunar south pole, contains water exceeding 5.6 ± 2.9 wt%, as determined from analyses of the regolith at the LCROSS impact site [35]. According to the LRO Mini-RF orbital radar report, an upper limit of 5 wt% to 10 wt% of water ice deposits (up to 30 vol.%) is present in the uppermost 1 m to 2 m section of the silicate regolith, and this is consistent with the observations obtained from the Clementine bistatic experiment [36].

This study focuses on extracting water ice from the floor of Shackleton Crater. For the considered case, the lunar regolith with the 8.4% water ice composition is projected to have uniaxial compressive strength ranging from 31 MPa to 43 MPa and relative density between 84 Dr% and 90 Dr% [37]. This regolith can be effectively excavated using surface bucket-drum excavation mining methods, similar to the methods employed for extracting moderate-strength limestones, sandstones, and shales [37,38]. According to Haruyama’s (2008) study, Shackleton Crater could be used for both surface and underground mining because its denser formation and the shallow layer of lunar dust above make accessing the water ice mineral easier [31]. Studies showed that it has a stable and uniform topography with very low rim height fluctuations, allowing for safer and easier landings [39,40]. The geological structures of Shackleton Crater and the landing site near it have been studied and taken from the LROC Quick Map as shown in Figure 1 [41].

### 3.2. Water Ice Extraction Process

In lunar craters at the south pole, ice exists in two forms: solid, where it binds regolith into rock-hard formations, and granular, as loose or loosely bound particles that can be excavated by scooping [42]. The regolith in Shackleton Crater contains between 5 wt% and 30 wt% ice, as indicated by the Lunar Propellant Outpost (LUPO) mission and Chandrayaan-1 M3 data [43,44]. In this study, we assume the regolith is granular and mixed with silicates [36]. Excavation robots will scoop the icy regolith, store it in block capsules, and transport it using hauler robots equipped with robotic arms. Water extraction will utilize low-power microwave heating, and its potential losses due to sublimation, system inefficiencies, and material transport must be considered [45]. A recent study by Liu et al. (2023) using drilling-based thermal extraction achieved only 80% water recovery, reinforcing the need for a conservative efficiency factor of 0.8 in our calculations [46]. Alternative extraction methods include reactive gas (hydrogen reduction, fluorination, and solid state and molten reduction), electrolysis reduction (molten regolith or molten salt electrolysis, e.g., the Fray, Farthing, and Chen (FFC) Cambridge process) and vapor-phase pyrolysis [19]. Solar energy from the rim of Shackleton Crater [47] will power the base station, processing plant, and maintenance site. It is worth noting that the specifics of the further processing of the regolith after water extraction are outside the scope of this study.

### 3.3. Constraints and Assumption

The development of the Lunarminer framework required addressing a number of technical limitations to ensure its efficiency and feasibility. One limitation is the disrupted communications within Shackleton Crater due to the lack of atmosphere and polarization effects [47,48], leading to the assumption that robots manage communications independently. In PSRs, limited visualization necessitates reliance on long-term planning and short-term management for navigation [49,50]. Energy efficiency is critical, as solar panel installation within the crater is not feasible, requiring optimized energy consumption and recharging strategies, similar to the approach for the tethered permanently shadowed region explorer (T-REX) [47]. Extreme temperatures and fine, loose regolith pose additional risks, such as traction loss and path deviations, necessitating path-tracking sensors and autonomous self-repair capabilities. These constraints underscore the need for a robust and adaptable framework to tackle the challenges of lunar water ice extraction.

## 4. Bio-Inspired Strategies and System Design

Swarm robotics leverages biomimicry, using behaviors of social animals like fish, ants, and bees to create small, collaborative robots capable of performing tasks in harsh environments, including in space [51,52]. Traditional large mining machinery would struggle to operate in lunar and Martian terrains due to mobility and maintenance challenges [53], as demonstrated by failures in rovers like Opportunity, Sojourner, and Curiosity [15,16,17]. Swarm robots mitigate these risks by distributing tasks across multiple agents, enhancing system reliability. The Lunarminer framework aims to optimize lunar exploration by integrating biomimetic principles from firefly synchronization and leafcutter ant division of labor.

### 4.1. Biomimicry in Swarm Robotics: Case Study

#### 4.1.1. Leafcutter Ants—Division of Labor

Leafcutter ants, especially from the Atta and Acromyrmex genera, utilize a division of labor in foraging that segregates tasks between leaf cutting and transporting, enhancing efficiency by reducing the energy spent climbing trees [54,55,56]. This behavior has inspired the development of energy-efficient swarm robotics systems [52]. Studies have implemented similar divisions of labor strategies in robotics, using designated roles to improve productivity and reduce task-switching [57,58,59]. Further simulations by Lee [60] and Tan et al. [61] demonstrate how dynamic task allocation in robots can significantly enhance performance in tasks, with Tan’s study showing a 27% increase in harvesting speed by dividing robots into specialized roles. Several case studies have applied leafcutter ants’ behaviors to swarm robotics [62,63,64].

#### 4.1.2. Fireflies—Synchronized Flashing Behavior

Fireflies in the Lampyridae family use synchronized bioluminescent flashing to attract mates, enhancing signal visibility over distances [65,66]. This behavior has inspired studies in swarm robotics for fault tolerance and recruitment tasks [52]. Christensen’s study applied firefly-like synchronization to detect failures in robotic swarms [67,68], while Prignano’s [69] and Wang’s studies [70,71] achieved faster synchronization with higher robot density using LEDs and cameras [72]. The discrete firefly algorithm (DFA) has been used in robotic mine clearance to improve task efficiency [73]. Maxseiner’s study demonstrated visible light communications in robotic swarms for reliable communication in challenging environments [74]. Tan’s [61] study showed that firefly-inspired flashing improved recruitment efficiency, increasing harvesting speed by 44% compared to the baseline model.

### 4.2. Lunarminer Bio-Inspired Concept

The Lunarminer framework is a biomimetic approach to extracting lunar water ice, leveraging the natural behaviors of leafcutter ants and fireflies to increase the efficiency, resilience, and adaptability of a robotic swarm at Shackleton Crater, as shown in Figure 2.

#### 4.2.1. Leafcutter Ants: Division of Labor and Task Allocation

The Lunarminer framework replicates the division of labor observed in leafcutter ant colonies, where ants specialize in tasks such as cutting leaves, gathering food, and transporting it to the nest. In these colonies, leaves are often cut and dropped at a cache, from which another group of ants collects and transports them to the nest [56]. Similarly, the Lunarminer framework assigns robots to specialized tasks like mapping the lunar surface, extracting water ice deposits, and transporting the extracted materials. To address the challenge of long-distance transport between the mine site and the processing plant, the framework mirrors the ants’ strategy by splitting the transportation task; namely, the robots first move ore blocks to a central hub near the mining site (analogous to the leaf cache), from where another group of robots transports the ore to the processing plant (analogous to the nest). This strategy reduces travel distances, conserves energy, and improves overall mining efficiency by effectively adapting the ants’ natural behavior to meet the demands of lunar mining.

#### 4.2.2. Firefly Bioluminescence: Recruitment Task and Fault-Tolerance Protocol

The Lunarminer framework draws inspiration from the bioluminescent communication of fireflies, which use light signals to navigate and coordinate in darkness. This is particularly crucial for lunar exploration in PSRs like Shackleton Crater, where sunlight is absent, rendering traditional visual aids ineffective [47]. In this framework, optical beacons and LiDAR technologies are employed for recruitment tasks, providing precise navigation and positioning by serving as fixed reference points. These beacons help the operating robots coordinate their movements and accurately locate mining sites in low-light environments. Additionally, the framework incorporates a recovery protocol modeled after the synchronous flashing behavior of fireflies [67,68]. When a robot experiences mechanical failure or runs out of energy, it emits a fault signal, prompting its replacement. This protocol ensures the resilience, robustness, reliability, and automation of the robotic swarm, allowing for continuous operation even if a robot fails.

### 4.3. Applicability of Other Social Animal Behavaiors

Apart from the coordinated behavior of fireflies and the division of labor among leafcutter ants, there exist other social animals that display actions that could potentially motivate different methods for extracting water ice from the moon. Honeybees, for instance, demonstrate sophisticated collective decision-making and resource allocation behaviors during foraging [75], which may be advantageous for robotic swarms looking to maximize resource extraction and minimize energy consumption [61]. Because atmospheric signals are impractical in hostile environments like the lunar vacuum, their reliance on a central communication scheme may prove challenging to implement. Similarly, large groups of social fish also exhibit smooth locomotion, which could be useful for group navigation in challenging lunar terrain [75]. Robots may move more effectively in challenging terrain, if they can stay close to one another without colliding. However, the application of fish swarms to multi-stage mining tasks like excavation, transportation, and resource exploration are limited, due to their lack of task specialization. These tasks require a more organized delegation of responsibilities. After analyzing these various tactics, we conclude that although each has its own advantages, the fault tolerance and synchronization of fireflies along with the division of labor and task specialization of leafcutter ants offer the most practical solution to the lunar mining problem. Leafcutter ants’ highly organized task management is especially helpful for organizing various phases of lunar mining and the firefly-inspired synchronization ensures robust fault tolerance, which is a necessary component for autonomous operation in such a remote location.

### 4.4. Swarm Robotic System Development

Transporting materials into space is prohibitively expensive, making the development of swarm robotic systems, which require large numbers of robots, a critical challenge. The Lunarminer framework addresses this by carefully optimizing the number of robots necessary for effective water ice resource extraction on the Moon. The specifications for these robots are based on NASA’s regolith advanced surface systems operation robot (RASSOR) 2.0, a planetary excavator with a technology readiness level (TRL) of 4. RASSOR 2.0 is 1.93 m long, 0.85 m wide, 0.43 m high, and weighs 67 kg with a 1410 Whr lithium battery [76], and with a power consumption of 4 W per kg of regolith excavation. It has a standard moving speed of 20 cm/s and a top speed of 56.5 cm/s [14,77]. The RASSOR 2.0 computer-aided design is illustrated in Figure 3 [14].

Based on the parameters of RASSOR 2.0, the Lunarminer framework defines the use of four explorer robots in the swarm design. Each of these robots has a movement speed of 0.72 km/h (20 cm/s), a sensor range of 10 m, and an 80% availability rate, enabling them to cover 0.46 km^2^ per day. This setup allows the entire 32.7 km^2^ floor of Shackleton Crater to be fully explored within 72 Earth days.

For excavation tasks, the framework specifies the deployment of two excavator robots, each capable of excavating 2.7 metric tons of regolith per day. This output can yield 1232 kg of oxygen through 45% molten salt electrolytic oxygen extraction [14,78]. Combined, these two excavator robots can produce approximately 2465 kg of oxygen daily, sufficient for 2770 people based on NASA’s estimate of 0.89 kg of oxygen consumption per astronaut per day [79]. To ensure continuous operation and to maximize efficiency, the Lunarminer framework incorporates a terrestrial mining method, where two hauler robots follow each excavator to maintain continuous frozen regolith harvesting and loading without downtime. Additionally, four hauler robots are included to maximize operational efficiency, while two transporter robots reduce transport times by moving the ore from the mine to the processing plant, collecting the ore deposits midway from the central hub [19].

## 5. Virtual Environment and Simulation

### 5.1. Virtual Lunar Environment Development

The proposed Lunarminer simulations were conducted in a ROS running Ubuntu Linux version 18.04 with a built-in Gazebo system and graphical user interface (GUI) for controlling and monitoring water ice extraction in a virtual lunar environment. The entire system was run on an HP Zen2 G5 TWR system with Intel(R) Core (TM) i7-10700 CPU, 16 GB RAM, and a 64-bit operating system. Built in ROS’s unified robot description format (URDF), the robot GUI was developed to be 0.15 m tall and 0.1 m long, and to weigh 60 kg. It was also equipped with a battery, advanced LiDAR sensors, automatic control systems, and decision-making algorithms. ROS functions such as Rviz and OpenCV were used to visualize specific robots and sensor data in the swarm to evaluate the effectiveness of the swarm robots for lunar water ice extraction and to lay the foundation towards practical implementation of the proposed autonomous lunar mining system.

A simulated virtual lunar environment was developed to mine water ice deposits, with a focus on mining over large areas rather than increasing the depth of excavation. This approach was based on LRO Mini-RF orbital radar reports that water ice deposits are located within the top 1 m to 2 m, eliminating the need for deeper excavation [36]. According to NASA Innovative Advanced Concepts (NIAC), when water vapor enters the Moon’s PSR from sources such as comet impacts, interplanetary dust, or space plasma interactions, it freezes in the coldest surface layers, forming a water vapor barrier that prevents deeper migration. This results in the accumulation of pure ice on the surface of the regolith, where fine powders of pure ice mix with soil during surface disturbance, which is supported by Hurley’s observations and models [80], and further supports excavation on the surface only, rather than in the subsurface. In addition, the limited power supply on the Moon requires optimizing energy consumption for surface operations. The microrobots designed for this mission further emphasize the need for energy efficiency, making deep excavation less feasible and less cost effective for the reasons stated above.

In addition, water ice deposits were found on the floor of Shackleton Crater, where the wall areas were not considered due to the need for advanced techniques such as grabbing or hanging that consume more energy and increase mining complexity. Therefore, the floor of Shackleton Crater was chosen for this case study. A virtual lunar environment of the floor of Shackleton Crater was simulated in ROS, with a flat gray surface (6.6 m × 6.6 m) representing the entire mining area. In the simulation, black circles represent collection points, yellow squares represent maintenance points, green squares represent base stations, blue squares represent processing points, green areas represent mining points, and blue areas represent transportation points. Robot teams are distinguished by different colors: four orange explorer robots are used for exploration, two green excavator robots are used for mining, four yellow hauler robots are used for collection, and two blue transporter robots are used for transporting minerals. Due to the limitations of ROS, 15 robots have been used to form a team, as shown in Figure 4.

### 5.2. Lunarminer Mining Lifecycle

The Lunarminer framework aims to propose a cutting-edge, bio-inspired approach to lunar water ice extraction, employing swarm robotics to navigate and operate in the challenging lunar environment. The framework’s mining lifecycle is governed by a finite state machine (FSM), which orchestrates the coordinated actions of the robots through each phase of the mining process. The framework is broken down into three key phases: resource prospecting and localization, mineral excavation and transportation, and maintenance and sustainability, as shown in Figure 5.

#### 5.2.1. Resource Prospecting and Localization

The resource prospecting and localization phase initiates the mining lifecycle, where exploration robots are deployed to explore the lunar surface, particularly targeting regions like Shackleton Crater. These robots are equipped with advanced LiDAR sensors for precise mapping and navigation, as well as specialized instruments such as neutron spectrometers and ground-penetrating radar (GPR) to detect and confirm the presence of water ice deposits beneath the lunar regolith. To effectively navigate the harsh and GPS-devoid lunar environment, the framework employs an advanced strip search strategy integrated with a piecewise tracking function:P=f(L−x)+p
where *L* represents the total length of the search path, *x* is the distance traveled by the robot along the path, and *p* is an offset to correct any deviations from the intended trajectory. This function ensures that exploration robots maintain a linear search pattern, preventing trajectory deviations and achieving precise area coverage. The adaptive navigation strategy is inspired by the bioluminescent behavior of fireflies, where the exploration robots mimic the light attraction behavior to guide their navigation in permanently shadowed regions of the lunar surface.

After detecting the water ice deposits, the exploration robot employs behavioral algorithms inspired by fireflies to strategically place light beacons, facilitating the recruitment of other robots for subsequent missions. This approach is similar to the recruitment strategies observed in DFA algorithms [73]. These beacons serve as recruitment signals for excavator and hauler robots, guiding them to the identified sites. The placement and influence of these beacons are governed by the following recruitment equations:$$R\left(x,y,t\right)=\sum_{i=1}^{n}\left(I\left(x_{i},y_{i}\right)\times e^{-\frac{D\left(x_{i},y_{i}\right)}{D_{max}}}\times S\left(t\right)\right)$$
where *R* (*x*, *y*, *t*) represents the recruitment potential at a given location and time, *I* (*x**i*, *y**i*) is the intensity of the light beacon, *D* (*x**i*, *y**i*) is the distance from the robot to the beacon, *D*max is the maximum effective recruitment distance, and *S*(*t*) is the temporal signal strength of the beacon. This model ensures that excavator and hauler robots are efficiently guided to the light beacon location for the subsequent phases of the operation. The simulation of this process is presented in Figure 6.

#### 5.2.2. Mineral Excavation and Transportation

During the excavation and transportation phases, the Lunarminer framework leverages firefly-inspired behavioral algorithms to enable the autonomous decision-making of the excavator robot. Similar to how fireflies are attracted to brighter light sources, as discussed in Fister et al. (2013) and demonstrated in the case studies [61,73], these robots determine their target locations depending on the intensity and proximity of the light beacons. The mathematical model for this behavior is as follows:$$TP\left(x,y\right)=\gamma \times \beta \left(D(x_{b},y_{b})\right)\times \frac{I(x_{b},y_{b})}{D{\left(x_{b},y_{b}\right)}^{2}}\times \frac{\left(x_{b}-x_{r},y_{b}-y_{r}\right)}{\sqrt{{\left(x_{b}-x_{r}\right)}^{2}+{\left(y_{b}-y_{r}\right)}^{2}}}$$
where *T**P* (*x*, *y*) is the movement vector towards the target position, and $\beta \left(D(x_{b},y_{b})\right)$ is the attractiveness function, which typically decreases with distance. The common form is *β*(*D*) = *β*_0_ × *e* – *m**D*^2^, where *β*_0_ is the attractiveness at distance *D* = 0, and *m* is a constant controlling the rate of decrease in attractiveness with distance. *I*(*x**b*, *y**b*) represents the intensity of the light beacon at coordinates (*x**b*, *y**b*). *D*(*x**b*, *y**b*) is the distance between the robot’s current position (*x**r*, *y**r*) and the beacon at (*x**b*, *y**b*), and *γ* is a constant that determines the speed of movement. This equation models the movement of an excavator robot toward the most favorable excavation site, and this movement is affected by the strength of the beacon and its distance from the robot’s current position.

The excavation process begins with the excavator robot rotating its bucket drum to dig the frozen regolith from the ground, storing it in ore block capsules for subsequent transportation. The hauler robot’s positioning system is seamlessly integrated into the excavation and collection phases, allowing for continuous mining and loading. Hauler robots, equipped with ore block sensors, locate the extracted water ice blocks and transport them to the designated collection site. To optimize this process, a division of labor mechanism inspired by leafcutter ants is employed, as observed in Labella’s studies [57,58]. The hauler robots are divided into two groups in a 50:50 ratio. The first group transports ore blocks from the mine site to the central hub, while the second group, known as transporter robots, carries the ore blocks from the central hub to the processing plant. This 50:50 split was selected based on the results of Labella [57,58] and earlier research by Tan et al. [61], indicating that in comparable task allocation scenarios a 50:50 division of labor works best. These earlier results imply that a 50:50 ratio is ideal for balanced resource transport tasks. This sequential handover system shortens the overall transportation distance and significantly enhances the efficiency and speed of material transportation to the processing plant. While one hauler robot collects the extracted water ice blocks, the other positions itself for subsequent loading, ensuring an efficient and uninterrupted workflow. The simulation of this process is presented in Figure 7.

It is worth noting that other ratios have not been investigated in this study. With the aim to further improve the performance of the proposed framework, future research will examine various ratios under operating conditions unique to the Moon.

#### 5.2.3. Maintenance and Sustainability

The maintenance and sustainability phase ensures the long-term viability of mining operations by implementing fault-tolerance mechanisms inspired by the synchronized flashing behavior of fireflies. When the robot malfunctions or runs out of energy, the LED light on top of the robot emits a red-light signal. The replacement robot detects this signal and activates the fault-tolerance protocol. This approach is based on methods presented in earlier studies [67,68,69,70,71]. The process is controlled by comprehensive fault-tolerance and task replacement algorithms:$$T_{r}=\frac{S_{i}}{D_{r}}\times \left(\theta \times \left(1-\frac{E_{i}}{E_{min}}\right)+\emptyset \times M_{i}\right)$$

Among them, *T**r* is the task replacement signal strength detected by the replacement robot *r*, *S**i* is the status indicator light of the robot *i* (1 means there is a fault, and the red light is on; 0 means running), and *D**r* is the distance between the faulty robot *i* and the replacement robot *r*. The value *η* is the scaling factor to adjust the sensitivity of the replacement robot to the signal, *E**i* is the current energy level of the faulty robot *i*, and *E**min* is the minimum operating energy threshold for the robot to operate. *M**i* is the fault indicator of robot *i* (1 if the robot is faulty and 0 if the robot is running), and *θ* and *ϕ* are weighting factors that balance the impact of energy consumption and fault conditions. Once a red signal is detected, a replacement robot moves to the location of the faulty robot and takes over its tasks. The malfunctioning robot automatically returns to the repair site for repairs. The system ensures continuous operations and enhances the resilience and sustainability of the mining process by minimizing downtime and maintaining operational efficiency. The simulation of this process is presented in Figure 8.

## 6. Results and Discussion

The simulation results offer a thorough assessment of the Lunarminer framework, specifically evaluating its navigation capabilities, material handling efficiency, and swarm automation processes. The space mining operation via the Lunarminer framework is illustrated in Figure 9.

### 6.1. Simulation Outcomes

The Lunarminer framework was tested in a simulated Gazebo lunar environment to investigate the operational coverage and exploration efficiency of four exploration robots. The primary objective was to evaluate the speed, coverage, and effectiveness of the search strategies employed when surveying and extracting water ice within Shackleton Crater. Operating at a standard speed of 14 cm/s, each exploration robot successfully covered 6.6 m in 46 s. This is equivalent to covering an area of 32.7 m^2^ in 4.6 min, or about 0.011 km^2^ of exploration space per day. Based on simulations, it was estimated that it would take nine Earth years for the four exploration robots to fully survey the 36 km^2^ floor of Shackleton Crater. This estimate does not consider potential downtime, or the time required to replace a robot, suggesting that while the framework is effective, further optimization will be required to accelerate the exploration process.

The water ice extraction capabilities of the Lunarminer framework were evaluated at five mining sites, each extracting four regolith blocks with a total weight of 320 kg per operating cycle. To accelerate the process, exploration started at X = 0, and the mining was confined to the area of 3.3 m × 6.6 m. The simulation set the extraction time per regolith block to 2 s to focus on transportation efficiency, applied a 20% failure rate to mirror the terrestrial mining industry-standard of 80% equipment utilization, and included a failure protocol. The simulation was tested 10 times, showing that it took an average of 23 min to deliver 20 regolith blocks from the mine site to the processing plant. Although the simulation employed a rapid 2 s extraction rate, under realistic conditions, the RASSOR 2.0 excavator extracts 80 kg (one regolith block) in 42 min [19]. This means that extracting 20 regolith blocks would take 830 min for excavation and 23 min for transportation. This is equivalent to extracting about 33 blocks of regolith, or 2640 kg of regolith, per Earth day. With a water ice concentration of 5.6 ± 2.9 wt% and a conversion efficiency of 80%, the daily water production is estimated to be 118 L. According to NASA’s Human Integrated Design Manual, this daily water production could support up to eighteen crew members, as sustainable space habitats with regenerative life support systems require 6.47 kg of water per crew member per day [79].

### 6.2. Comparative Analysis

The Lunarminer framework has demonstrated superior water production efficiency compared to both the mobile in situ water extractor (MISWE) and the system reported by Battsengel et al. (2023) [81]. According to Zacny et al. (2012), a single MISWE rover, equipped with a 1 m deep auger and a 5 cm diameter drill, recovered 0.2 L of water per hour, totaling 4.8 L per day [18]. Deploying 12 MISWE units, equivalent to the number of robots in the Lunarminer framework, would yield 57.6 L per day. In contrast, the Lunarminer framework nearly doubles this output, producing 118 L per day. Additionally, the research of Battsengel et al. [81], involving 10 robots extracting icy regolith in Shoemaker Crater at a speed of 152 m/h across two shifts per day, achieved an annual extraction of 146 tons of regolith, translating to an extraction of 22.4 L of water per day. Scaling this to 12 robots would result in an extraction of 26.9 L per day, which is significantly lower than the 118 L per day achieved by the Lunarminer framework. The comparison of the Lunarminer framework with alternative technologies is shown in Table 1.

The comparative analysis of energy usage and extraction efficiency between swarm robots with DoL and non-DoL strategies in lunar mining operations has demonstrated the significant benefits of task allocation inspired by leafcutter ants. The simulations were conducted across three different setups, each with five mine sites containing two, four, and six ore blocks, respectively. Two strategies were tested: (1) a 100:0 ratio without DoL, where six haulers delivered ore directly to the processing plant, and (2) a 50:50 ratio with DoL, where four haulers transported ore to a central hub, and two transporters completed the delivery. The energy consumption trends were derived from assumptions based on NASA’s RASSOR 2.0 bucket-drum excavator [19] and a parametric review by Just et al. (2020) [82]. Key estimates included 320 W for excavating an 80 kg ore block [14,83,84], 90 W for fully loaded haulage per minute [84,85], 10 W for unloaded haulage per minute, and 5 W for mineral detection. The light beacon power was excluded to focus on the overall energy trend from the DoL strategy implementation. The analysis of extraction efficiency and energy usage for DoL and non-DoL strategies is illustrated in Figure 10a and Figure 10b, respectively.

As shown in Figure 10a, the DoL strategy has reduced the extraction time by 25%, 30%, and 40% for 10, 20, and 30 ore block scenarios, respectively. This reduction was due to optimized workload distribution, reduced travel distances, and minimized idle times. Energy consumption data have further supported the efficiency of the DoL strategy. As depicted in Figure 10b, the total energy usage in the DoL setup peaked at around 1600 watts between 6 and 8 min and then declined, stabilizing at about 600 watts by the end of the operation. In contrast, the non-DoL setup peaked at 1800 watts and remained high for longer, reflecting the energy-intensive nature of hauling ore directly over greater distances. Additionally, the energy graph shows that the DoL strategy has resulted in a more stable energy distribution compared to the non-DoL strategy, where energy usage was more erratic and prolonged. This stability in energy distribution has indicated a more controlled and efficient use of resources, reduced the overall energy demand, and enhanced operational efficiency. The primary factor driving these differences was the reduced travel distance in the DoL setup, where specialized roles for haulers and transporters have optimized the energy usage. In contrast, the non-DoL approach, requiring robots to cover the full distance from mine to processing plant, resulted in prolonged energy consumption, with energy peaks lasting nearly twice as long. These results have highlighted the DoL strategy’s effectiveness in improving operational efficiency, shortening extraction times by up to 9 min, and reducing total energy consumption by approximately 31%, making it particularly advantageous for lunar mining where energy efficiency is crucial.

The comparative analysis of fault tolerance and system robustness within the Lunarminer framework was evaluated by implementing a protocol inspired by the synchronized flashing behavior of fireflies. The evaluation was carried out under three distinct scenarios: (1) a normal scenario with no failures, where all robots operated at full efficiency; (2) a failure scenario with a 50% failure rate, where half the robots malfunctioned after 12 min of operation; and (3) a recovery scenario, where a recovery protocol was implemented to replace malfunctioning robots, also beginning at the 12 min setup. The simulations were conducted on a standardized mine setup consisting of five sites, each containing four ore blocks, resulting in a total of twenty ore blocks to be harvested. The analysis of recovery efficiency is illustrated in Figure 11.

As Figure 11 shows, the results highlight the critical role of the recovery protocol in maintaining system robustness, automation, and resilience within the Lunarminer framework. In the normal scenario, all 20 ore blocks were harvested in approximately 19 min, reflecting full operational efficiency. In contrast, the failure scenario extended harvesting time to 35 min, an 84% increase due to the reduced number of operational robots, which slowed the ore collection rate. The recovery scenario completed the harvesting process in 22 min, showing a 15.8% increase compared to normal operations but significantly outperforming the failure scenario. The recovery protocol has proved to be effective by quickly replacing malfunctioning robots, with production temporarily stalling between the 13th and 16th minutes during robot replacement in 12 min. After the recovery, the system rapidly restored efficient operation, as indicated by the sharp decline in remaining ore blocks after 16 min. Although production efficiency briefly dropped during the recovery period, the system quickly returned to a collection rate close to normal operations. In contrast, the failure scenario experienced a steady decline in efficiency, demonstrating the recovery protocol’s importance in improving operational efficiency by 37.1% compared to the failure scenario. This analysis has underscored the recovery protocol’s ability to significantly mitigate the impact of robot failures and to maintain performance levels close to optimal conditions.

### 6.3. Environmental Condition Analysis

Temperatures in the PSR region will continue to drop as lunar mining advances, and under extreme circumstances, the lunar surface may see significant temperature fluctuation from up to 127 degrees Celsius during the day to −173 degrees Celsius at night [19]. Robotic systems are severely affected by these variations, particularly in terms of battery efficiency and electronics performance. The robotic swarm is distributed, which offers robustness even though the current Lunarminer framework does not specifically include sophisticated thermal management systems. Should an extreme environment cause a failure of robots, the loss of one robot will not have an impact on the system, enabling the remaining functional units to carry on with their work. To further improve resilience in harsh environments, future research should examine modifications to thermal management. Moreover, visual navigation is challenging in the PSR, due to limited sunlight. The Lunarminer robots use non-visual sensors like radar and LiDAR for obstacle detection and terrain mapping. The robot swarm uses synchronized light signals inspired by fireflies to coordinate the movements of the robots in the absence of light. With this method, robots can continue to carry out their missions and communicate effectively even in areas that are dark.

### 6.4. System Performance Within Current Robot Limitations

Since the ROS simulation environment has limitations, the current paper considers a swarm of 15 robots. For this number of robots, the system is optimized for task distribution and fault tolerance. Although the performance of larger swarms is not tested in this study, it is expected that the decentralized architecture of the framework, where each robot independently plans tasks and communicates with the rest of the swarm robots, would allow the system to scale effectively. Future research will try to evaluate this scalability with more robots, investigating the effects of more robots on overall performance, task coordination, and energy efficiency.

### 6.5. Lunarminer Framework Integration

The proposed Lunarminer framework integrates collective behaviors inspired by nature, namely the division of labor observed in leafcutter ants’ foraging behavior and the synchronized flashing of fireflies, to create an autonomous, efficient, and resilient swarm robotic system. Drawing from the swarm robotics behavior classification [52,75,86], the Lunarminer framework can be categorized as shown in Figure 12.

The Lunarminer framework classifies the proposed swarm robotics behaviors by incorporating strategies inspired by both leafcutter ants and fireflies. Through the division of labor modeled after leafcutter ants, the system has achieved spatial organization via self-assembly, object clustering, and assembly, with navigation for collective transportation, while also facilitating decision-making for consensus, task allocation, and collective perception. Additionally, the synchronized flashing behavior of fireflies has contributed to the system’s self-healing capabilities, enhanced consensus-based decision-making, synchronization, collective perception, and fault detection. This approach also supports navigation through collective exploration, coordinated motion, and collective localization.

The Lunarminer framework has incorporated nature-inspired strategies to optimize various aspects of lunar mining operations [75]. The division of labor, modeled after leafcutter ants, has enhanced transportation efficiency, leading to reduced times in mine planning and operation. The synchronized flashing behavior of fireflies has facilitated navigation and ore detection in dark regions, which is essential for exploration and operational efficiency. Additionally, this behavior has contributed to the system’s self-healing capabilities, thereby improving reliability during mining activities. While mine closure and rehabilitation are not within the scope of this framework, the focus was placed on minimizing environmental impact throughout the mining process. The application of these nature-inspired behaviors across the mining lifecycle is presented in Table 2.

### 6.6. Validation of the Lunarminer Framework

The performance metrics used to evaluate the efficacy of the Lunarminer framework have been defined through the quantitative outcomes of the system’s operation in a simulated lunar environment. The realistic environment of Shackleton Crater was replicated using the robot operating system (ROS). Important metrics like water extraction rate (181 L per day), energy efficiency (31% reduction), time efficiency (up to 40% reduction in extraction time), and fault tolerance (20% robot failure tolerance) are summarized in Table 3.

By comparing the framework’s performance directly to that of current technologies, these metrics show how resilient, scalable, and operationally efficient the Lunarminer system is. The findings in Table 3 confirm that the system can function well on the Moon, outperforming competing technologies in terms of extraction rate, energy efficiency, and system resilience. The simulations also show that the Lunarminer framework is a reliable option for lunar resource extraction, because it can continue to function effectively even in the face of difficult circumstances like a 20% robot failure rate.

## 7. Conclusions

The Lunarminer framework was developed to test the hypothesis that biomimetic swarm robotics, inspired by the division of labor in leafcutter ants and the synchronized flashing behavior of fireflies, may significantly enhance the automation, efficiency, reliability, and sustainability of lunar water ice extraction. The simulations conducted within Shackleton Crater demonstrated that the proposed framework may effectively improve operations through optimized task allocation, increased energy efficiency, and enhanced system resilience, thereby supporting the initial hypothesis proposed by the authors. Specifically, the division of labor modeled after leafcutter ants may enable the robotic swarm to optimize ore transportation, resulting in a reduction in the extraction time by up to 40% and in energy consumption by approximately 31% in scenarios with high ore block quantities. Additionally, the firefly-inspired recovery protocols may enhance the fault tolerance of the Lunarminer system, maintaining near-optimal operational efficiency even in the presence of robot failures. The framework’s capability to produce up to 181 L of water from excavated regolith with an overall conversion efficiency of 0.8 allows for further validating its potential as a possible reliable solution for sustained lunar mining operations, ISRU, and supporting Moon habitation supplies for up to 18 crew members. These findings demonstrate the potential effectiveness of the Lunarminer framework in addressing the challenges of lunar resource extraction, particularly in permanently shadowed regions like Shackleton Crater. The integration of nature-inspired behaviors not only may allow for optimizing production and swarm operations but also may provide a scalable and resilient approach to ISRU for future space missions. Although the results of the simulation show real-world applications, the system will face other challenges like severe temperature fluctuation, lunar dust, and communication lags which should be resolved before deployment. In order to further validate the framework’s performance, physical testing under conditions that more closely resemble the lunar environment in facilities such as the EXTERRES laboratory will be essential. In this study, a cluster of only 15 robots was issued for simulation, because of the ROS version’s limitations. Future research should aim to overcome this constraint, because larger robot clusters might offer more profound understanding of the system’s robustness and scalability. By contributing to the development of sustained operations for long-term lunar exploration, these real-world validations will assist in converting the simulation results into useful applications for lunar mining. To assess how well this framework performs with larger clusters, future studies will investigate more sophisticated computing frameworks or distributed simulation environments. It is recommended that further research on optimizing the framework for large-scale deployment in space mining operations be conducted and the proposed strategies be further refined using rover systems and regolith thermal vacuum chambers (RTVACs) available at the EXTERRES laboratory at the University of Adelaide.

## Figures and Tables

**Figure 1 biomimetics-09-00680-f001:**
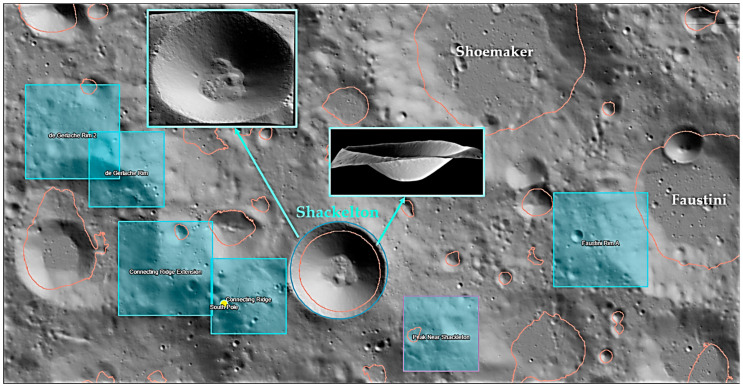
Landing sites near Shackleton Crater on the lunar south pole, marked with blue squares, and highlighting the geological formations near Shackleton Crater [41].

**Figure 2 biomimetics-09-00680-f002:**
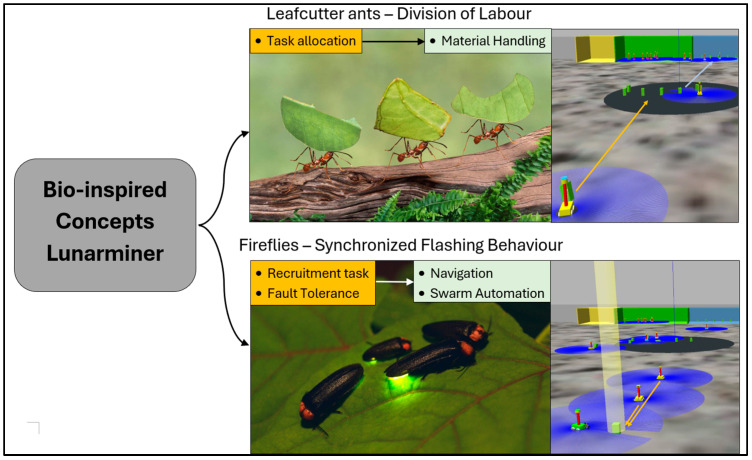
Bio-inspired design concepts for the Lunarminer framework. The orange boxes represent specific tasks, i.e. task allocation and material handling (inspired by leafcutter ants), and recruitment and fault tolerance (inspired by fireflies),which contribute to broader goals stated in the green boxes, such as efficient navigation, swarm automation, and resource handling.

**Figure 3 biomimetics-09-00680-f003:**
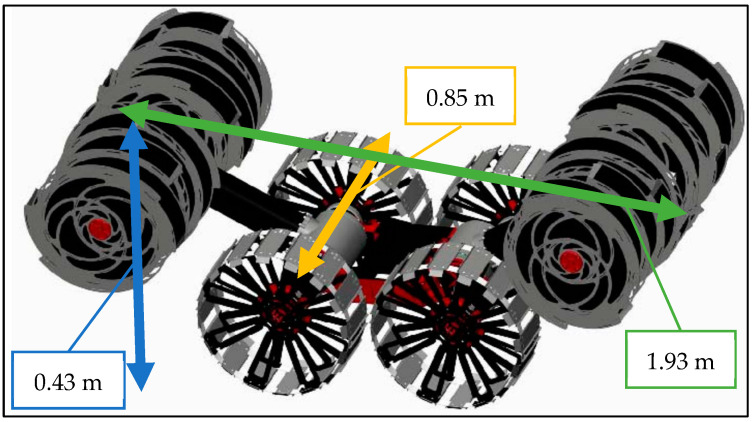
RASSOR 2.0 computer-aided design [14].

**Figure 4 biomimetics-09-00680-f004:**
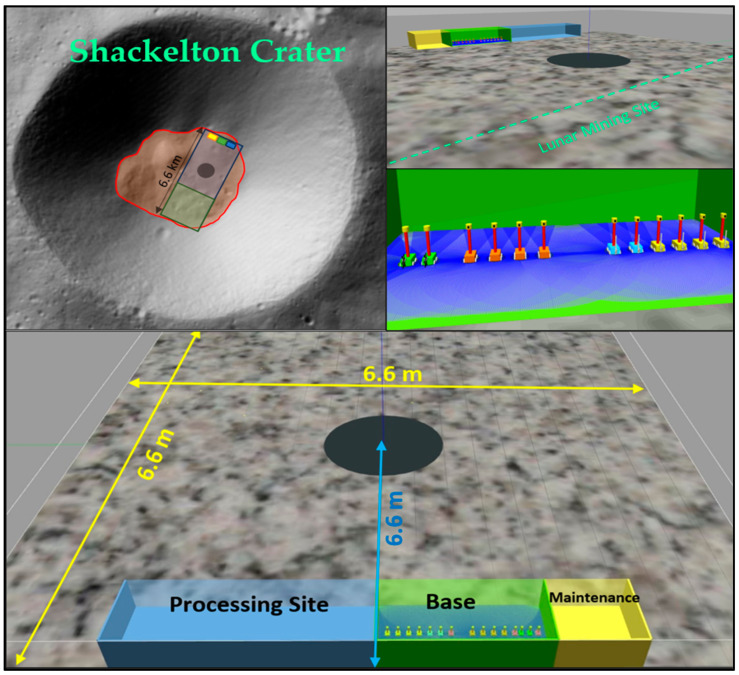
Simulated virtual lunar environment. ROS simulation of Shackleton Crater’s floor (gray areas) with a central hub for collection (black circles), maintenance (yellow squares), base stations (green squares), processing (blue squares), mining (green areas), and transportation (blue areas). The robot fleet includes 4 orange explorers, 2 green excavators, 4 yellow haulers, and 2 blue transporters.

**Figure 5 biomimetics-09-00680-f005:**
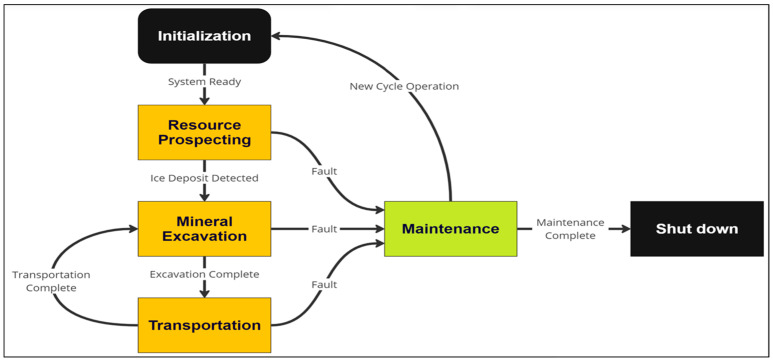
Lunarminer finite state machine.

**Figure 6 biomimetics-09-00680-f006:**
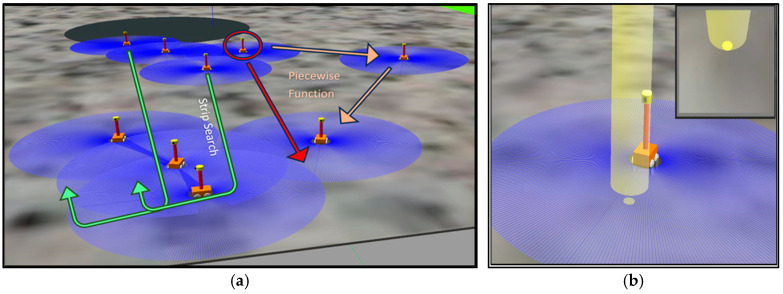
(**a**) Strip search and piecewise tracking function for resource prospecting. The green arrows indicate the strip search path, while the red arrow highlights a prioritized direction or specific target location within the search area. Blue-shaded areas represent zones covered by individual units as they scan for resources; (**b**) fireflies’ bioluminescent function inspired recruitment protocol, where light beacons are placed at ore locations to signal and attract other units to ore locations.

**Figure 7 biomimetics-09-00680-f007:**
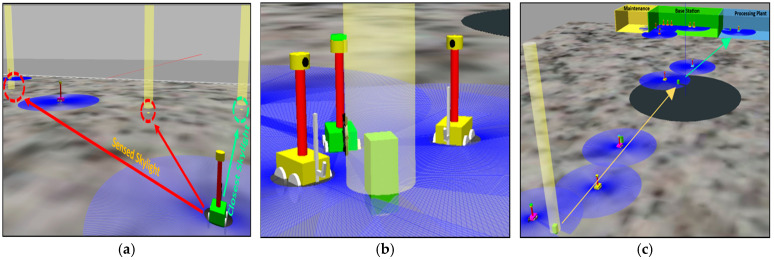
(**a**) Selection of the mining site based on light proximity and intensity, with red arrows indicating sensed skylight directions guiding site selection; (**b**) mining excavation process showing ore block detection and hauler positioning system, with light beacon areas representing operational zones.; and (**c**) division of labor in transporting ore blocks: yellow arrows indicate transport paths from the mine site to the central hub, and green arrows show paths from the central hub to the processing plant.

**Figure 8 biomimetics-09-00680-f008:**
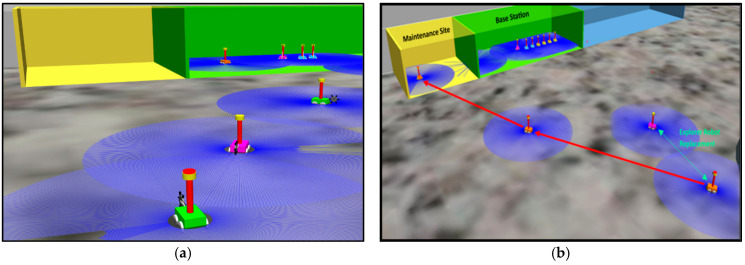
(**a**) A red-light signal emitted by a malfunctioning robot, inspired by the flashing behavior of fireflies, with blue shaded areas representing the communication range of each robot; (**b**) activation of the fault-tolerance protocol to replace the malfunctioning robot, indicated by red arrows guiding the replacement robot toward its target within the blue communication zones. The base station and maintenance site are shown in green and yellow, respectively, facilitating the coordination of the replacement process.

**Figure 9 biomimetics-09-00680-f009:**
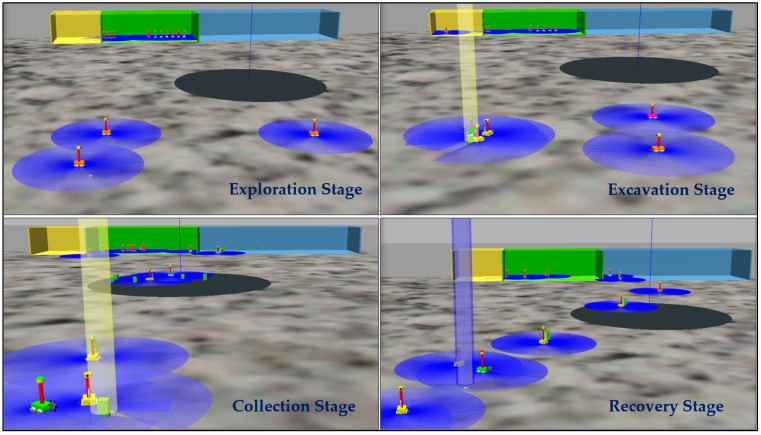
Highlights of various stages of the Lunarminer mining process from the exploration stage to the recovery stage.

**Figure 10 biomimetics-09-00680-f010:**
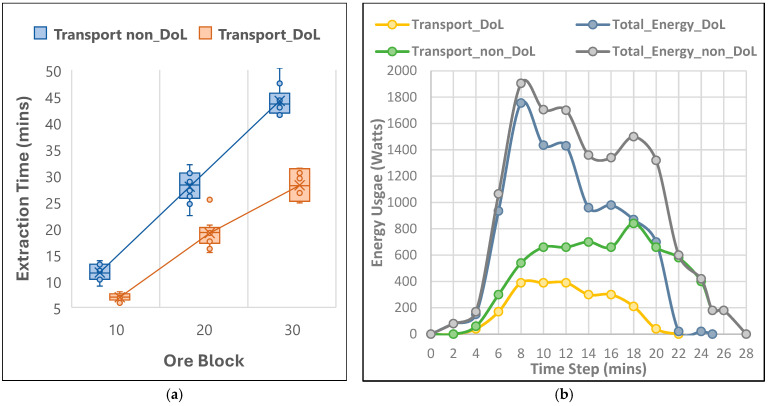
(**a**) Resource extraction time and (**b**) energy distribution across different scenarios.

**Figure 11 biomimetics-09-00680-f011:**
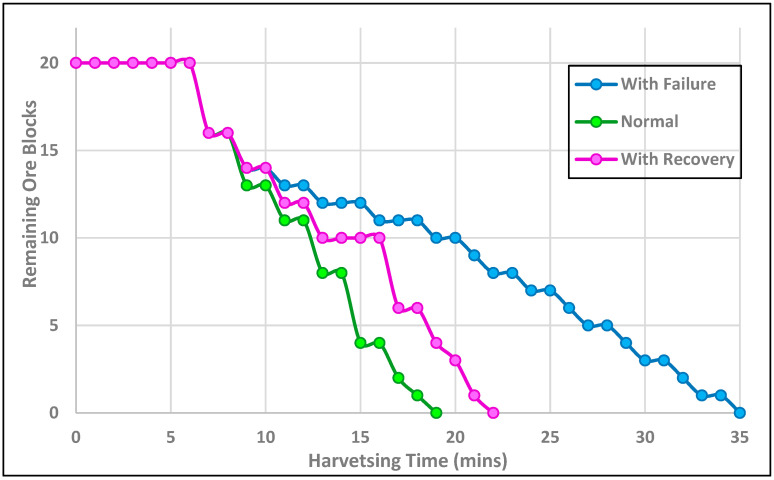
Fault tolerance and system robustness across three scenarios, i.e., normal, with failure, and with recovery settings.

**Figure 12 biomimetics-09-00680-f012:**
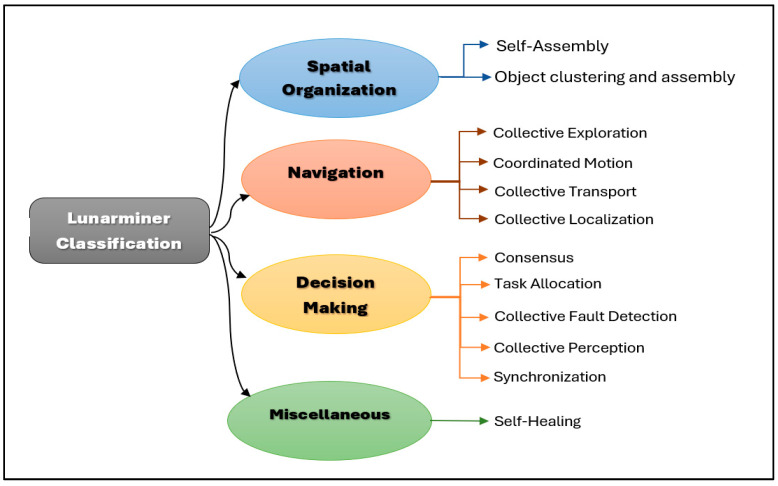
Lunarminer framework classification.

**Table 1 biomimetics-09-00680-t001:** Comparison of Lunarminer Framework with Alternative Technologies.

Technology	Water (L)/Regolith (kg) Extraction Rate	Energy Efficiency (Watts/Liter or Regolith)	Fault Tolerance	Scalability	Notes
Lunarminer Framework	181 L/day 2640 kg/day	4.2 watts/L	Up to 20% robot failure	High (Decentralized control)	Achieves superior water extraction with significant energy savings and fault tolerance.
RASSOR	800 kg/day	4.1 watts/kg	Limited mechanical resilience	Medium (Single unit)	Designed primarily for regolith excavation, it consumes more energy for large-scale excavation.
MISWE	4.8 L/day	Not specified	High energy consumption per unit	Low (Single unit, low tolerance)	Limited by low extraction rates, scaling to 12 units yields 57.6 L/day.

**Table 2 biomimetics-09-00680-t002:** Lunarminer Mining Lifecycle.

Mining Lifecycle Phase	Leafcutter Ants	Fireflies
Mine Exploration and Assessment	No	Yes
Mine Planning and Design	Yes	Yes
Mine Operation and Construction	Yes	Yes

**Table 3 biomimetics-09-00680-t003:** Lunarminer Framework Performance Metrics for Each Mining Phase.

Mining Phase	Metric	Value	Notes
Mine Exploration	Area covered per robot per Earth day	0.46 km^2^	A total of 15 robots were used to fully explore Shackelton crater’s 32.7 km^2^ floor in 72 Earth days.
Regolith Excavation	Total blocks excavated per day	33 blocks (2640 kg of regolith)	Daily excavation of 2640 kg of regolith.
	Time per block excavation	42 min	Time to excavate one block of 80 kg regolith.
Water Extraction	Water extraction rate	181 L/day	Water produced per day from excavated regolith with 5.6 wt% water ice composition.
Energy Efficiency	Energy savings	31% energy reduction	Energy savings in high ore block quantities.
Operational Time Efficiency	Time reduction in extraction	Up to 40% reduction	Time savings achieved through optimized task allocation.
System Resilience	Robot failure tolerance	Up to 20% failure rate	The system operates efficiently even with a 20% robot failure rate.

## Data Availability

The original contributions presented in the study are included in the article, further inquiries can be directed to the corresponding authors.

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
