# Peer review of "Lunarminer Framework for Nature-Inspired Swarm Robotics in Lunar Water Ice Extraction"

_biomimetics, 2024, doi:10.3390/biomimetics9110680_

Round 1
Reviewer 1 Report
Comments and Suggestions for Authors
Dear Authors, please find my comments, questions, and suggestions below.
MDPI Author Layout Style Guide recommends the following:
Most abbreviated phrases should be written in full the first time that they are used, with the abbreviation in brackets, for example “small angle X-ray scattering (SAXS)”.
The abbreviation LCROSS appears for first time in the line 64 without explanation, which is given further in line 98. The explanation of some abbreviations is given several times: ISRU is explained in lines 28, 55, 552; ROS is explained in lines 85, 246; PSRS is explained in lines 62, 140, 206. Other abbreviations, namely LSII, UCS, and VLC were introduced once and were not used further in the text of the manuscript. Please check all abbreviations and ensure they are in accordance with MDPI's Style Guidelines.
Your Lunarminer framework is inspired by the division of labour in leafcutter ants and the synchronized flashing of fireflies. And in lines 150-152 of Article you mentioned other social animals. Have you evaluated their potential applicability to the lunar water ice extraction problem? What advantages and disadvantages have you identified in alternative approaches? You could describe it in more details, what might be helpful for some readers and your colleagues.
In lines 218-223 you wrote:
Transporting materials into space is prohibitively expensive, making the development of swarm robotic systems, which require large numbers of robots, a critical challenge. The Lunarminer framework addresses this by carefully optimizing the number of robots necessary for effective resource extraction on the Moon. The specifications for these robots are based on NASA's Regolith Advanced Surface Systems Operation Robot (RASSOR) 2.0, a planetary excavator with a Technology Readiness Level (TRL) of 4 [73].
I think it would be appropriate to include a picture of RASSOR in the article, as well as its dimensions, since you previously showed leafcutter ants and fireflies in Figure 2. Providing a visual representation of the robot will make it easier for readers to understand and visualize the information presented.
In lines 282-283 you wrote:
Due to the limitations of ROS, we consider using 15 robots to form a team, as shown in Figure 3.
Is it possible to overcome this limitation and conduct research with more than 15 robots? If so, you could add information about this in the Conclusion section in the part about future work if any are planned.
In lines 356-358 you wrote:
The hauler robots are divided into two groups in a 50/50 ratio. The first group transports ore blocks from the mine site to a central hub, while the second group, known as transporter robots, carries the ore blocks from the central hub to the processing plant.
In your research, did you find that the ratio of 50/50 was always optimal, or did you experiment with different ratios until you found the most effective one? Did you search for the most optimal ratio under different parameters of the robot swarm workflow simulation?
I think it would be more clear to use +/- or yes/no instead of the current color coding in Table 1. The color marking is not so clear.
In general, I think your article deserves publication in Biomimetics Journal after minor revision.
Reviewer 2 Report
Comments and Suggestions for Authors
This paper presents the Lunarminer Framework including a swarm robotic system inspired by biomimetic behaviors to optimize lunar water-ice extraction. The simulation results show an increased operational efficiency by reducing task allocation time by 40% and an energy consumption by 31%.
1- Introduction section should include comparative analysis with existing methods and technologies to clearly differentiate the Lunarminer framework. Some recent references should be added for it.
2- Related works should include direct comparison of technologies and methods related to in-situ resource utilization to the Lunarminer approach. The reader needs to know how the framework improves upon or differs from them.
3- The paper needs the analysis of various environmental conditions such as temperature fluctuations, different lighting conditions and the system performance under more robots.
4- Table 1 could be improved with more quantitative data like performance metrics for each phase.
5- A table should be added for comparison with alternative technologies of available results.
6- There is no method/sign about the validation of effectiveness of the Lunarminer framework.
7- In conclusion, give some comments about the translation into real-world applications of the simulation results.
Round 2
Reviewer 2 Report
Comments and Suggestions for Authors
The final improved paper is sufficient.